# Seminal Extracellular Vesicles and Their Involvement in Male (In)Fertility: A Systematic Review

**DOI:** 10.3390/ijms24054818

**Published:** 2023-03-02

**Authors:** Ana Parra, Lorena Padilla, Xiomara Lucas, Heriberto Rodriguez-Martinez, Isabel Barranco, Jordi Roca

**Affiliations:** 1Department of Medicine & Animal Surgery, Faculty of Veterinary Medicine, International Excellence Campus for Higher Education and Research “Campus Mare Nostrum”, University of Murcia, 30100 Murcia, Spain; 2Biotechnology of Animal & Human Reproduction (TechnoSperm), Institute of Food and Agricultural Technology, University of Girona, 17003 Girona, Spain; 3Department of Biomedical & Clinical Sciences (BKV), BKH/Obstetrics & Gynaecology, Faculty of Medicine and Health Sciences, Linköping University, 58185 Linköping, Sweden; 4Department of Veterinary Medical Sciences, University of Bologna, Ozzano dell’Emilia, 40064 Bologna, Italy

**Keywords:** seminal plasma, extracellular vesicles, fertility, infertility, proteins, sncRNAs

## Abstract

Seminal plasma contains numerous extracellular vesicles (sEVs). Since sEVs are apparently involved in male (in)fertility, this systematic review focused on studies specifically investigating such relationship. Embase, PubMed, and Scopus databases were searched up to 31 December 2022, primarily identifying a total of 1440 articles. After processing for screening and eligibility, 305 studies were selected as they focused on sEVs, and 42 of them were considered eligible because they included the word fertility or a related word such as infertility, subfertility, fertilization, and recurrent pregnancy loss in the title, objective(s), and/or keywords. Only nine of them met the inclusion criteria, namely (a) conducting experiments aimed at associating sEVs with fertility concerns and (b) isolating and adequately characterizing sEVs. Six studies were conducted on humans, two on laboratory animals, and one on livestock. The studies highlighted some sEV molecules, specifically proteins and small non-coding RNAs, that showed differences between fertile and subfertile or infertile males. The content of sEVs was also related to sperm fertilizing capacity, embryo development, and implantation. Bioinformatic analysis revealed that several of the highlighted sEV fertility-related proteins would be cross-linked to each other and involved in biological pathways related to (i) EV release and loading and (ii) plasma membrane organization.

## 1. Introduction

Seminal plasma (SP) is the accumulated mixture of fluids produced by the testis, epididymis, ductus deferens, and, primarily, by the accessory sex glands surrounding spermatozoa during and after ejaculation [1]. Seminal plasma has a heterogenous composition that includes many bioactive molecules involved in regulating the main functions of spermatozoa and the functional activity of cells of the female genital tract once it is deposited during copulation or artificial insemination (AI) [1]. In fact, some SP bioactive molecules are essential for the display of sperm motility and for sperm capacitation, as well as providing a favorable uterine immune environment for sperm transit and, beyond fertilization, for embryo development and implantation. Therefore, the search for biomarkers of male fertility among bioactive molecules of SP is a timely research subject [2]. These bioactive molecules may circulate either freely in the SP or packaged within extracellular vesicles (EVs), where they would remain safe from the several natural inhibitors present in the SP, such as proteases or nucleases.

Extracellular vesicles are lipid bilayer nanovesicles usually ranging in size from 30 to 1000 nm in diameter, carrying a wide range of molecules, including proteins, nucleic acids, metabolites, and lipids [3]. They are generated by most functional cells of the organism that release them to the extracellular milieu either from endosome-derived multivesicular bodies (small EVs) or by direct budding from the plasma membrane (large EVs) [4]. The EVs circulate freely in any body fluid until they bind to target cells either neighboring or distant from their point of origin, which, after transferring their cargo, could reprogram their functional behavior [5]. Thereby, EVs play an essential role in cell-to-cell communication, and they are involved in many pathologies, including cancer, immunity, neurodegenerative disorders, and physiological reproductive processes such as embryo implantation and placenta development [6,7,8]. Therefore, active molecules carried by EVs are revealed as potential non-invasive biomarkers of many body (dys)functions [9,10], including reproductive ones such as polycystic ovary syndrome and premature ovarian failure [11,12].

Seminal plasma contains a comparatively higher amount of EVs than blood or cerebrospinal fluid [13], and although they were among the first organic fluid EVs isolated and characterized, they remain to date among the least explored [14]. Seminal EVs (sEVs) are secreted by functional cells of the testis, epididymis, vas deferens, and male accessory glands, in particular, prostate and vesicular glands [15]. Like EVs circulating in other body fluids, those in seminal plasma conform to a heterogeneous population displaying conspicuous differences in size, shape, electrodensity, and composition [8,16]. Seminal EVs bind and exchange active molecules with mature sperm- and endometrial epithelial cells and are thus involved in multiple reproductive events [8,15]. Accordingly, sEVs would be involved in the regulation of sperm motility, capacitation, and acrosome reaction, all functional sperm attributes essential for fertilization. Moreover, the cargo load of sEVs differs between normozoospermic and non-normozoospermic males [17]. Seminal EVs would also facilitate the safe transit of spermatozoa in the female genital tract by regulating the uterine immune response [18]. Collectively, these findings would indicate that sEVs may be involved in fertility and that some biomolecules encapsulated into sEVs could be postulated as biomarkers of male fertility. However, there are currently no molecules recognized as biomarkers of male fertility among those encapsulated in sEVs. To determine how far science has advanced in this field, the present systematic review aimed to summarize existing published evidence on the relation of sEVs on fertility and the identification of candidate EV molecules that would act as biomarkers of fertility or infertility.

## 2. Materials and Methods

The systematic review was conducted according to the PICO framework and following the guidelines outlined in the Preferred Reporting Items for Systematic Reviews and Meta-Analyses (PRISMA 2020) statement [19] (Figure 1A). The systematic review has been sent to PROSPERO for registration (ID 325967).

### 2.1. Selection of Peer-Reviewed and Published Studies

The search of the available literature was conducted using Embase “https://www.embase.com/ (last accessed on 1 February 2023)”, PubMed “https://pubmed.ncbi.nlm.nih.gov/ (last accessed on 31 January 2023)”, and Scopus “https://www.scopus.com/ (last accessed on 31 January 2023)” databases. The three databases were reviewed from inception through 31 December 2022. The search was conducted by two of the signatory reviewers (A.P. and J.R). working independently but following the same search strategy. Disagreements were resolved with input from two additional signatory reviewers (L.P. and I.B.). The searched terms included combinations of the keywords epididymosome, prostasome, extracellular vesicle, exosome, microvesicle, semen, seminal plasma, epididymis, ejaculate, sperm, and spermatozoa, which were entered in the search box following the tags or syntactic guidelines of each search engine. The terms queried in each database are shown in Appendix A. The records of all identified articles were then screened for eligible articles.

### 2.2. Screening, Eligibility, and Inclusion/Exclusion Criteria

The criteria for screening, eligibility, and inclusion applied in this systematic review are depicted in the flowchart of Figure 1B. These procedural steps in the review research were carried out by the signatory reviewers A.P. and J.R., and any inconsistencies or discrepancies that arose were resolved through debates with two other signatory reviewers, I.B. and L.P. The set of references identified from the searches in the three databases was screened to select only articles written in English and reporting experimental studies. Accordingly, reviews, books and books chapters, abstracts of conference presentations, letters, commentary articles, replies, errata, and editorials were all excluded. The selected records from each database were collected in a single database, and duplicates were removed. Then, the selected articles were examined to choose those that focused on sEVs. The first eligibility criterion was that the articles should include sEV or related words such as epididymosome, prostasome, exosome in the title, abstract, and/or keywords. Next, the selected articles were subjected to a second eligibility criterion to choose those that related sEVs with fertility issues. For this, the eligibility criterion was that articles had to include the word fertility or a related word such as infertility, subfertility, fertilization, and recurrent pregnancy loss (RPL) in the title, objective/s, and/or keywords. Finally, to define which articles would be included and which would be excluded, the full text of the eligible articles was reviewed. Only articles that (a) performed experiments aimed at associating EVs with fertility problems and (b) properly isolated and characterized EVs using ISEV-recommended methods [20] were included. Proper isolation and characterization of EVs are critical methodological steps that provide robustness in evaluating the results obtained in EV research studies.

### 2.3. Data Extraction

Information from each eligible article was extracted separately by two signatory reviewers (A.P. and J.R.). The signatory reviewers, I.B. and L.P., collaborated to resolve any inconsistencies or discrepancies by reviewing the original data of the articles under debate. The extraction process was carried out in two sequential steps. Firstly, the following items were recorded to select the articles focused on the sEVs: (1) title; (2) reference data, including authors, the country where the study was conducted, journal, and year of publication; (3) authors keywords; and (4) research topic, including species. Secondly, to select the articles relating sEVs to fertility traits from among the articles that focused on sEVs, the following items were also recorded: (1) objective/s, (2) procedures for EV isolation and characterization, (3) experimental design, and (4) main findings.

### 2.4. Bioinformatic Analysis

The proteins and microRNAs (miR) identified in the included studies were searched in ExoCarta “http://exocarta.org (last accessed on 1 February 2023)” and Vesiclepedia “http://microvesicles.org (last accessed on 1 February 2023)”, two databases recording the proteins, RNAs, and lipids identified in EVs [21,22], to verify whether identified in the sEVs and, if so, if their role in male reproductive function was known.

The Search Tool for the Retrieval of Interacting Genes/Proteins (STRING; “https://string-db.org/ (last accessed on 10 February 2023)” was used to identify possible interactions between the proteins highlighted in the included articles. The Cytoscape software with the ClueGO plugin “http://www.cytoscape.org/ (last accessed on 10 February 2023)” was used to create functional pathway networks of the proteins identified in the included articles. To do this, the integrated Gene Ontology (GO) terms and Kyoto Encyclopedia of Genes and Genomes (KEGG) pathways integrated into ClueGO were used. The following ClueGO filters were used: GO tree levels 2–4 (first level = 0), minimum number of genes, 2, minimum percentage of genes, 4, and a kappa score of 0.4. The resulting networks were manually reordered and modified to remove some redundant and unnecessary terms. A similar bioinformatic analysis of the highlighted miRNAs was not possible because they were not accurately identified in the included articles.

## 3. Results

### 3.1. Search Report

The screening and eligibility processes led to the inclusion of a total of 305 references focused on sEVs (Appendix A). The first article was published in 1982, and since then, the number of articles published has exponentially grown. In total, 12 articles were published in the 1980s, 46 in the 1990s, 63 in the 2000s, 109 in the 2010s, and 75 from 1 January 2020 to 31 December 2022. The articles have been published in 130 scientific journals, highlighting Andrology (including the former Journal of Andrology and International Journal of Andrology) with 24 published articles, The Prostate with 19 articles, and Biology of Reproduction with 18 articles. The research was conducted in laboratories of 28 countries, with Sweden (61 articles), the United States (51 articles), China (39 articles), Italy (31 articles), Canada (24 articles), Spain (14 articles), Germany (10 articles), and the United Kingdom (10 articles) standing out (Figure 2A). The most active research centers have been Uppsala University Hospital, Sweden, with 55 published articles; the Faculté de Medicine de la Université Laval, Québec, Canada, with 23 articles; and the Istituto di Biochimica e Chimica Medica, Università di Perugia, Italy, with 18 articles. The research published in the 305 eligible articles was mainly conducted in humans (190/305, 62.3%), distantly followed by laboratory animals (rodents, 43/305, 14.1%), cattle (22/305, 7.2%), and swine (21/305, 6.9%). Seminal EVs from other species, either mammalian (horse, dog, chicken, cat, and sheep) or non-mammalian (reptile, amphibian, fish, and fly), were also investigated, although to a lesser extent (Figure 2B). The main topics addressed in the eligible articles were the characterization of sEVs (121/305, 39.7%), distantly followed by those addressing the relationship of sEVs to sperm function (48/305, 15.7%), the interaction of sEVs with sperm (44/305, 14.4%), those relating sEVs to fertility/infertility (42/305, 13.8%), and those relating sEVs to diseases (40/305, 13.1%), mainly prostate cancer and those caused by sexually transmitted viruses. Other research topics were also addressed, but to a lesser extent, including the functional performance of EVs, their biogenesis, isolation, usefulness as biomarkers, their involvement in male reproductive disorders and sperm maturation, and their functional role in the female reproductive tract (Figure 2C). Many of the articles addressed two or more research topics, most notably relating to EV characterization and fertility, EV characterization and sperm interaction, EV characterization and sperm function, and male reproductive disorders and fertility.

A total of 42 articles were considered eligible because they comprised either the word “fertility” or other related words in the title, objective, and/or keywords (Appendix A). However, once read, 18 of them (42.9%) were excluded because they did not conduct studies or experiments relating sEVs to fertility features, and other 15 (35.7%) were also excluded because they did not report isolation and/or sEV characterization (Appendix A). Thus, only nine articles (9/42, 21.4%) were finally included (Table 1), whose research was carried out in laboratories in eight different countries, and each of them was published in a different scientific journal.

The distribution of the articles finally included is shown in Figure 3, distributed by species and research topics. Research in six of the nine articles was conducted in humans (66.7%), two in laboratory animals (22.2%), and one in livestock (11.1%). The research in humans focused mainly on the involvement of sEVs in sperm disorders causing infertility (4/6, 66.7%). The other two focused on the involvement of sEVs in sperm fertilization capacity (1/6, 16.7%) and RPL (1/6, 16.7%). Those conducted in laboratory animals focused on evaluating the involvement of sEVs in aging-related male infertility (one article) and the relevance of specific sEV-proteins on fertility success (one article). The article reporting research carried out in a livestock species examined the phenotypic and compositional differences of sEVs between fertile and subfertile roosters. Eight of the nine articles explored differences in molecules loaded in sEVs between fertile and infertile males. Specifically, six articles focused on the protein load of EVs (66.7%), either by analyzing the proteomic profile using high-throughput techniques or by focusing on individual proteins, and two articles focused on small non-coding RNAs (sncRNAs) (22.2%). The other article analyzed the differences between fertile and subfertile males in the phenotypic characteristics of sEVs and in their ability to fuse and internalize into spermatozoa.

### 3.2. Seminal EVs vs. Male (In)Fertility

Table 2 summarizes the reference data, sample details, study questions, and main results of the nine finally included articles. Four articles compared molecules loaded into sEVs from infertile men, displaying different sperm disorders (non-normozoospermic men), with those from fertile men showing normozoospermia. Two of these articles focused on proteins. Garcia-Rodriguez et al. [26] found that normozoospermic men displayed a unique set of proteins expressed in sEVs (Ras-related protein Rab-22A and 26S protease regulatory subunit 10B) while those of non-normozoospermic men displayed another set including charged multivesicular body protein 2b, exportin-1, and 40S ribosomal protein S19. Vickram et al. [30] found that clusterin, a 52 kDa protein with a key role in sperm capacitation and motility, was in lower amounts in the sEVs of non-normozoospermic men compared to normozoospermic men. The other two articles focused on sncRNAs. Abu-Halima et al. [23] identified a panel of microRNAs (miRs) either overexpressed (miR-765, miR-1275, and miR-1299) or underexpressed (miR-30b, miR-20a, miR-148a, miR-26b, and miR-15a) in sEVs of oligoasthenozoospermic men. More recently, Hong et al. [27] found four piRNAs (piR-1207, piR-2107, piR-5937, and piR-5939) underexpressed in the sEVs of asthenozoospermic men.

Three studies related sEVs with sperm fertilization, embryo development, and pregnancy success. Mei et al. [29], using an in vitro fertilization experiment, demonstrated that a reduced expression of galectin-3 in human sEVs would be behind low fertilization rates. Jena et al. [28] compared protein profiles of sEVs of fertile men with those of couples of women with RPL, finding 177 deregulated proteins in sEVs of the male partners of RPL-affected women. Some of the underexpressed proteins were DNA-related, and their under-expression would lead to defective sperm chromatin packaging and histone deletion, which, resulting in inappropriate expression of paternal genes, would consequently result in abnormal embryo development. Other sEV-deregulated proteins were related to defective maternal immune response to paternal antigens, leading to impaired placenta decidualization. The other article, conducted in a mouse animal model and focused on the Arrestin Domain Containing 4 (Arrdc4) protein that is involved in EV biogenesis and release, showed that sperm from Arrdc4-/- mice were unable to fertilize oocytes [25].

The other two studies focused on the phenotypic characterization of sEVs. Wang et al. [31], also using a mouse animal model, demonstrated that age-related changes in the size and content of sEVs would impair embryo implantation by affecting the immune environment of the female uterus. Cordeiro et al. [24], the only article reporting research in livestock that compared the size and content of sEVs from fertile and subfertile roosters, found that sEVs from fertile roosters were smaller, contained more heat shock protein 90-alpha (HSP90A), and had greater sperm fusion and internalization capacity than those from subfertile roosters.

### 3.3. Bioinformatic Analysis

Figure 4A lists the 21 sEV molecules that the above-reported studies have found related to male fertility issues. Specifically, they were proteins (9/21, 42.9%), miRs (8/21, 38.1%), and piRNAs (4/21, 19%). All proteins and miRs highlighted in the included articles were listed in Exocarta and/or Vesiclepedia. In addition, one of the highlighted proteins, HSP90AA1, is among the top-ten most identified EV proteins. However, database listings only include some of these molecules as identified in sEVs, specifically, the proteins CLU, HSP90AA1, LGALS3, and the miR-26a. None of the listed articles that had identified these proteins and miR in sEVs are from the included studies. This is surprising considering the databases became operational in 2009 (ExoCarta) and 2012 (Vesiclepedia), and all the included articles were published from 2016 onwards.

The sEV proteins highlighted in the included studies were analyzed using STRING and Cytoscape to determine possible relationships among them and to discover related functional pathways. STRING generated a single protein–protein interaction network including five genes (Figure 4B). The resulting protein–protein interaction network was analyzed with the ClueGo plugin of the Cytoscape software, revealing that the encoded proteins appear involved in two pathways, namely: chaperone-mediated autophagy and positive regulation of inclusion body assembly (Figure 4C).

## 4. Discussion

### 4.1. Search Report

A total of 305 articles evaluated sEVs until 31 December 2022, a relatively small number compared to the large number of articles focused on EVs circulating in other body fluids such as blood, cerebrospinal fluid, or urine [14]. Regarding sEV-articles, it is peculiar that the EVs present in SP of most species are commonly termed prostasomes, particularly in articles evaluating those of human SP. Prostasome indicates a prostate gland origin and would not be too appropriate to refer to all EVs of SP, as they may have sources other than the prostate gland. For instance, testes, ducti epididymides, deferent ducts, and vesicular glands are also able to generate and release EVs to SP [15]. Therefore, sEVs would be a more appropriate term as it encompasses all EVs of SP, regardless of their origin. This proposal was already suggested by Renneberg et al. [32], but unfortunately, it has not been widely implemented. Many of the 305 articles showed research focusing—as expected—on the phenotypic and compositional characteristics of sEVs, key steps for further understanding EV functional roles. Noteworthy were the few numbers of articles focusing on the biogenesis and isolation of sEVs. Looking at biogenesis, in addition to the two common mechanisms accepted for the secretion of EVs circulating in other body fluids, other releasing mechanisms are contemplated for sEVs [33], with apocrine secretion as the probable major mechanism behind [34]. Accordingly, further research—albeit challenging—is needed to reach more conclusive evidence.

In total, 42 of the 305 articles were initially considered eligible, and, upon close reading, 18 of them were excluded because they did not address experiments relating sEVs to fertility, despite including the word fertility or other related words in the title, objective, or keywords, a clearly misleading action. Authors should be more rigorous in their choice of article titles and/or keywords. An additional 15 articles were also excluded for not following ISEV standards for the isolation and/or characterization of EVs. This surprising finding is particularly striking in the five articles published from 2014 onwards since the first recommendations for the study of EVs were issued in that particular year (MISEV, 2014), clearly highlighting the need to isolate and correctly characterize EVs [35]. The characterization of EVs by multiple and complementary methods is mandatory to ensure that molecules identified as associated with EVs are truly associated with EVs and not with other particles or co-isolated materials [36]. These methodological shortcomings call into question the validity of the results obtained, as there is no evidence of the presence of EVs in the samples analyzed or whether only EVs were present in the samples. These considerations led to the exclusion of the articles.

### 4.2. Seminal EVs vs. Male (In)Fertility

The nine finally included articles reported that the content of sEVs differed between fertile males and those with fertility disorders suggesting sEVs and their contents are, somehow, involved in male fertility. This small number of articles also reflected the still very limited search for fertility biomarkers among the molecules loaded in sEVs, when compared to other body fluids such as blood, saliva, or urine, where biomarkers for the early diagnosis of several diseases, including some types of cancer, have been identified among the molecules loaded in EVs [37,38].

A review of the performed experiments and the results obtained in each of the nine eligible articles showed that four of them underwent large-scale’omics research using high-throughput techniques. Specifically, two carried out proteomics [25,33] and the other two transcriptomics [24,28]. All of them were timely performed and provided very valuable results. Yet they should nevertheless be considered exploratory studies, as they focused on the identification of sets of proteins or sncRNAs that were differentially expressed between sEVs from fertile and infertile males. Notable among these studies was the one conducted by Garcia-Rodriguez et al. [26], which found a set of proteins present only in sEVs from fertile men or infertile men. Specifically, the ras-related protein Rab-22A and the protease regulatory subunit 26S 10B were only in the sEVs of fertile men, and the charged multivesicular body protein 2b, exportin 1, and the ribosomal protein 40S S19 in the sEVs of men with fertility impairments associated with sperm disorders. Unfortunately, no subsequent studies were performed evaluating the suitability of these differentially expressed proteins or sncRNAs as potential biomarkers of fertility. Comprehensive characterization of the sEV proteome, transcriptome, lipidome, and metabolome are essential requirements for better understanding their functional role and identifying EV molecules that may be involved in molecular pathways related to sperm fertilization capacity and subsequent embryo development. These ´omic studies must, however, be followed by validations and evaluations of the usefulness of identified molecules as potential fertility biomarkers. Such studies, particularly the latter, are still pending.

Four articles evaluated the relationship of specific sEV molecules with fertility performance. Vickram et al. [30] highlighted clusterin, an important seminal plasma glycoprotein linked to sperm capacitation, oxidative stress, and immune regulation in the female reproductive tract [39]. Vickram et al. [30] found lower amounts of clusterin in SP and sEVs from infertile men, suggesting that quantification of clusterin from SP and/or sEVs could be a potential biomarker for the diagnosis of human male infertility. Mei et al. [29], using an in vitro fertilization experiment, demonstrated that a reduced expression of galectin-3 in human sEVs would be behind low fertilization rates. In this regard, it is important to remark that the galectin-3 protein of the sEVs would play an important immunoregulatory role in the female reproductive tract, facilitating implantation and embryonic development [40]. The study by Foot et al. [25] in a mouse model, focusing on Arrestin Domain Containing 4 (Arrdc4), a protein involved in EV biogenesis and release [41], showed that spermatozoa from Arrdc4-/- mice were unable to fertilize oocytes, which would be related to the fact that EV-Arrdc4 is important for sperm maturation [25]. Finally, the study by Cordeiro et al. [24] performed in chicken showed that the HSP90AA1 protein was expressed to a greater extent in the sEVs of fertile roosters than in those of subfertile roosters. In a previous study, this protein associated with sEVs was found in higher amounts in frozen-thawed spermatozoa from low-fertility bulls [42]. These apparently conflicting results alert us of the need for further studies on the role of this protein in male fertility.

Seminal plasma plays an important role in embryo development and implantation [1], a role that could be played by sEVs [43,44]. The study by Wang et al. [31] conducted with laboratory animals demonstrated that sEVs would be involved in embryo development, although it does not highlight which sEV molecules would be involved. On this matter, the above-mentioned proteomic study of Jena et al. [28] showed that sEVs would be involved in RPL. Taken together, these two articles would confirm the relevance of sEVs not only for early embryonic development but also for subsequent implantation.

Artificial breeding via AI is widely used in livestock, and its optimal performance requires the use of fertile breeding males. Unfortunately, it is relatively common to find sub-fertile males in AI centers, males that successfully overcome regular, apparently rigorous semen quality controls [45]. Use of semen AI-doses of these subfertile males causes reproductive and economic losses. Therefore, finding accurate biomarkers of male fertility remains an important as a challenge for the livestock industry. In this context, it is striking that there is only one article that investigated the relationship between sEVs and male fertility in livestock. The study of Cordeiro et al. [24] conducted in chicken demonstrated that the phenotypic characteristics of sEVs of fertile roosters were different from those of sub-fertile cocks. Although not providing conclusive results, this study showed that sEVs may be related to male subfertility, therefore encouraging further studies in other species of livestock.

### 4.3. Bioinformatic Analysis

At present, there are free EV databases listing EV cargo from data uploaded by researchers, providing essential background information for further research. Unfortunately, none of the included studies have uploaded their data to these databases. This shortcoming hinders future research on sEVs. Researchers are strongly encouraged to upload their EV data to these free databases to facilitate the advancement of knowledge on the composition and functionality of sEVs.

The included studies identified sEV molecules, mainly proteins, which were found in different amounts between fertile and infertile individuals. However, most of these studies did not reveal the biological and molecular pathways in which these proteins would be involved. STRING revealed interaction between some of the highlighted proteins, and Cytoscape analysis revealed that they would be involved in two pathways. Specifically, the Chaperone-mediated autophagy protein would be related to EV release and loading [46], and the positive regulation of inclusion body assembly protein would be related to plasma membrane organization [47], probably that of gametes. Unfortunately, bioinformatic analysis of miRs was not possible because the data provided by included articles were not sufficiently refined for accurate database analysis. Most of the highlighted miRs have two different sequences with different target genes, and the included articles did not indicate which sequence was predominant [48].

### 4.4. Strengths and Limitations

This is the first review to systematically examine published research linking EVs to male fertility. The PRISMA-inspired search strategy was intended to be as inclusive as possible by expanding the search to three renowned databases such as Embase, PubMed, and Scopus. Despite this, the review has identified surprisingly little research conducted to date on this subject, which contrasts with the already considerable and growing body of research focused on sEVs and the socioeconomic relevance of male infertility in both human and livestock species. The review has also surprisingly identified that most of the research studies published so far on this subject do not include a section displaying the characterization of EVs, which is currently mandatory. This circumstance limits the robustness of the results, a matter that must be fulfilled in future studies. The review also revealed a few sEV-bound proteins and sncRNAs as candidate biomarkers of (in)fertility, although they should be further evaluated.

The most important limitation of this review is the small number of studies included, jeopardizing solid conclusions. Moreover, the studies developed very different experimental designs, preventing any comparative or cross-analysis. Another limitation would be associated with the search strategy conducted, which, while aiming to be as inclusive as possible, does not guarantee that all the existing scientific articles on this subject were included, either because they were written in languages other than English or because they did not include the word fertility or another related word in the title, objective, or keywords.

### 4.5. Considerations for Future Studies

The nomenclature used to refer to the EVs of semen should be harmonized. The term prostasome should be avoided as a general defining name unless pointing out the source of origin, as it does not define all EVs present in semen, irrespective of the dominance of the prostate in humans. Using the term, sEVs would be a better option, being inclusive for all EVs in semen, regardless of their origin, biogenesis, and phenotypic characteristics.

To improve the robustness of the results, studies should clearly prove that samples under evaluation contain EVs free of contaminating particles or molecules. To this end, we suggest following the recommendations periodically provided by the ISEV. The latest published guide [20] provides excellent recommendations for the isolation of EVs and their characterization and for performing functional studies.

Existing studies in humans are based on single experiments or clinical trials with a small number of individuals/samples under evaluation, which undermines the robustness of the results. Therefore, it is essential to conduct studies that explore larger numbers of individuals/samples. Collaboration between institutions/research centers would be beneficial to do this since it would allow to analyze a greater number of individuals/samples, particularly for human clinical trials. Unfortunately, such collaborations have not been common in the studies carried out to date.

More studies in laboratory and livestock species would be desirable. These species allow trials with many individuals or samples, not only from semen donors but also from healthy breeding females, since semen from a single male is often used by AI to inseminate many females with a single ejaculate. Furthermore, in these species, in addition to fertile and infertile males, there are also subfertile males, a reproductive status that may facilitate a better understanding of the relationship of seminal EVs to fertility outcomes. 

## 5. Conclusions

The few existing research studies, the methodological limitations of many of them, and the few solid results obtained demonstrate that research focused on the association of sEVs with fertility is still in its infancy, and it remains, therefore, an open topic for future research. Despite these drawbacks, the studies so far available suggest that sEVs would be involved in male fertility. Evidence that the content of the sEVs varies between fertility-compromised non-normozoospermic males and fertile normozoospermic males would support this claim. Seminal EVs would be involved in the fertilizing capacity of the spermatozoa and even in the subsequent development and implantation of the embryo. Research is still limited to date and has not yielded sufficiently robust results to identify biomarkers of male fertility among the molecules loaded in sEVs.

## Figures and Tables

**Figure 1 ijms-24-04818-f001:**
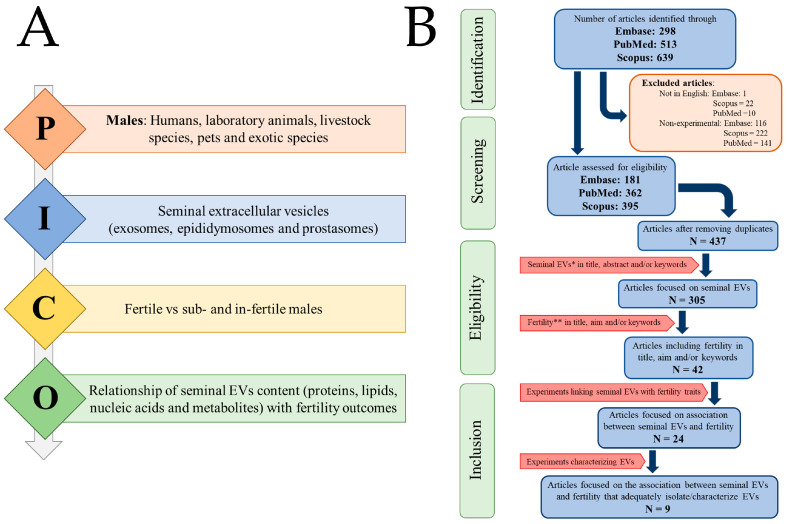
Flowchart showing the article selection process carried out following (**A**) the principle of PICO (P: Population; I: Intervention; C: Comparison; O: Outcome) and (**B**) the guidelines of PRISMA 2020. “N” indicates the number of items.

**Figure 2 ijms-24-04818-f002:**
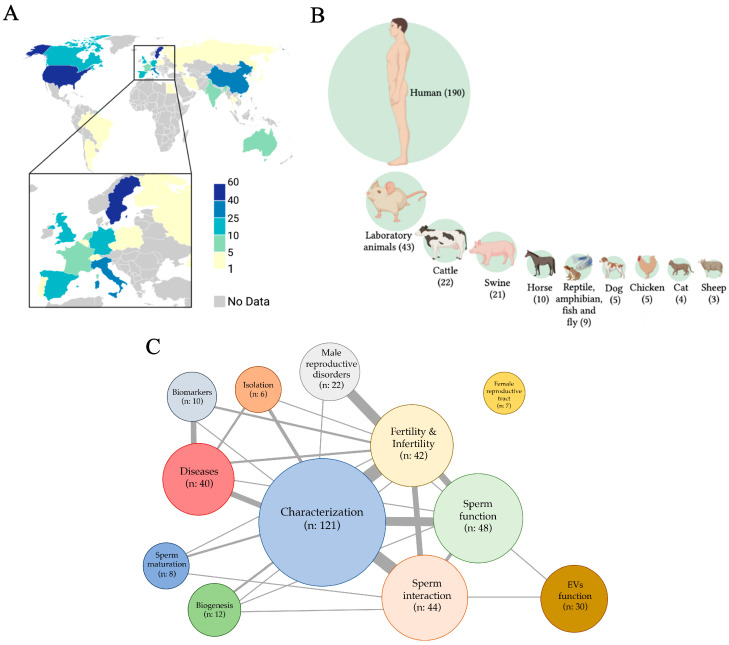
Research on seminal extracellular vesicles (sEVs). (**A**) Countries of research teams, (**B**) species, and (**C**) research topics. The drawing was created using the software of BioRender.com.

**Figure 3 ijms-24-04818-f003:**
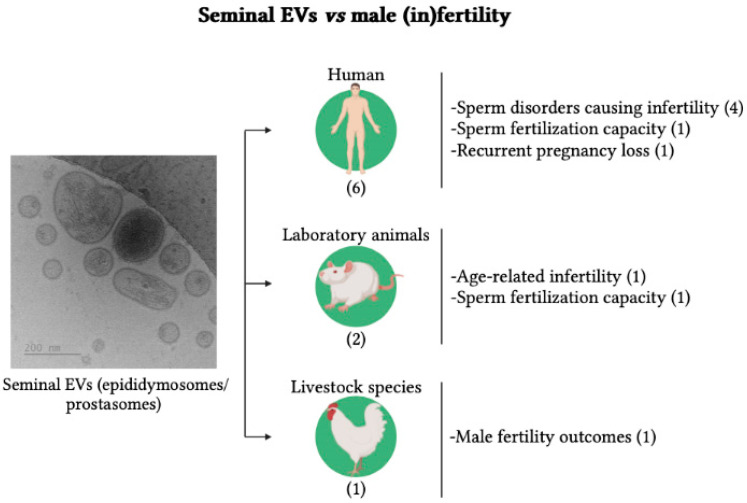
Seminal extracellular vesicles (EVs) showing heterogeneity in size, shape, and electrodensity, and the articles included in the systematic review that relate them to male fertility distributed according to species and research topic. The numbers indicate the number of articles. The cryo-electron microscope image is from the archive of the research group (pig seminal EVs) and was generated at the exosome laboratory of CIC bioGUNE (Vizcaya, Spain). The drawing was created on BioRender.com.

**Figure 4 ijms-24-04818-f004:**
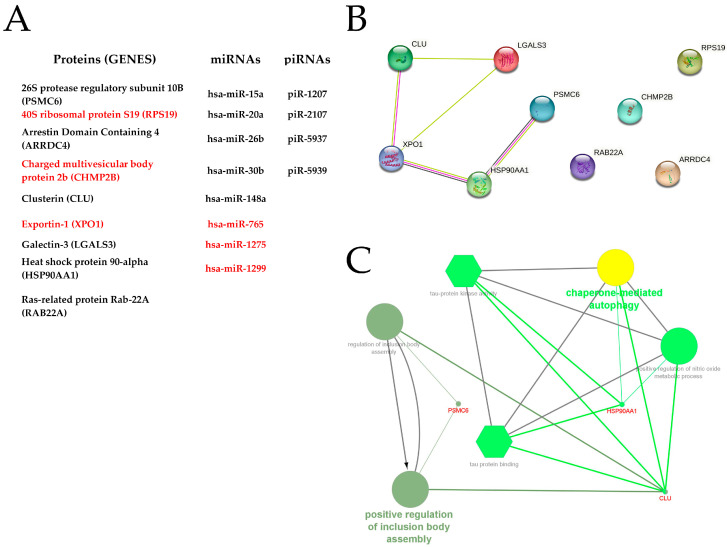
Bioinformatic analysis of the seminal EV molecules highlighted in the eligible articles. (**A**) list of the highlighted molecules. Molecules in black were positively related to fertility, and molecules in red were negatively related to fertility. (**B**) Protein–protein network generated by STRING. Circles depict genes, lines show interactions between the gene-encoded proteins, and line colors indicate evidence of protein–protein interactions. (**C**) Cytoscape analysis showing the biological processes (circles) and molecular functions (hexagons) of the target genes of the protein network generated by STRING. The drawing was created using the software BioRender.com.

**Table 1 ijms-24-04818-t001:** Procedures used for the isolation and characterization of seminal extracellular vesicles (sEVs) in the nine articles included in the systematic review.

Authors	Species	Isolation Method	Characterization
Abu-Halima et al. (2016) [23]	Human	UC + gradient UC	EV-specific markers (WB: CD63, CD81, CD9 and HSP70)
Cordeiro et al. (2021) [24]	Chicken	UC (×2)	Morphology (TEM), EV-specific markers (WB: ANXA5, HSP90A, VCP, and PDCD6IP)
Foot et al. (2021) [25]	Mouse	UC	Size distribution and concentration (NTA), EV-specific markers (WB: CD81, Annexin A1, and B-actin)
García-Rodríguez et al. (2018) [26]	Human	UC (×2)	Total protein quantification (BCA assay), size distribution and concentration (NTA), morphology (TEM)
Hong et al. (2021) [27]	Human	UC	Size distribution and concentration (NTA), Morphology (TEM), EV-specific markers (WB: CD63 and TSG101)
Jena et al. (2020) [28]	Human	Filtration + UC	Size distribution (DLS), morphology (SEM), EV-specific markers (WB: CD9 and CD81)
Mei et al. (2019) [29]	Human	Filtration	Morphology (TEM), EV-specific markers (WB: CD13)
Vickram et al. (2020) [30]	Human	UC + SEC	Size distribution (DLS), morphology (SEM), composition assay (EDX)
Wang et al. (2021) [31]	Mouse	UC + filtration	Size distribution and concentration (NTA), morphology (TEM), EV-specific markers (WB: CD63 and TSG101)

Abbreviations: ANXA5 (Annexin A5), BCA assay (Bichinchoninic acid assay), DLS (dynamic light scattering), EDX (energy dispersive X), sEV (seminal extracellular vesicles), HSP90A (Heat Shock Protein 90 Alpha), NTA (nanoparticle tracking analysis), PDCD6IP (Programmed Cell Death 6 Interacting Protein), SEC (size-exclusion chromatography) SEM (scanning electron microscope), TEM (transmission electron microscope), TSG101 (Tumor Susceptibility Gene 101), UC (ultracentrifugation), VCP (Valosin-containing protein), WB (Western blot).

**Table 2 ijms-24-04818-t002:** Sample details, study questions, and main results of the nine articles included in the systematic review.

Authors	Country	Species	Sample Details	Study Question	Main Results
Abu-Halima et al. (2016) [23]	Germany	Human	Semen from normo- (*n*:12) and oligoastheno-zoospermic men (*n*:12)	Whether altered miRNA expression profiles of EVs were related to fertility	Eight miRNAs were differentially expressed between oligoastheno- and normo-zoospermic men
Cordeiro et al. (2021) [24]	France	Chicken	Seminal plasma from fertile (*n*:7) and subfertile (*n*:6) males	Whether sEVs from fertile and subfertile chickens showed differences in their characteristics and sperm fusion capacity	sEVs from fertile and subfertile roosters showed differences in size, protein composition, and sperm fusion capacity
Foot et al. (2021) [25]	Australia	Mouse	Samples from cauda epididymis and vas deferens (*n*:156)	Whether Arrdc4-/-, a protein involved in EV biogenesis, improves the fertilizing capacity of sperm	Supplementation of Arrdc4-/- spermatozoa with seminal EVs improved their fertilizing capacity
García-Rodríguez et al. (2018) [26]	Spain	Human	Semen from normo- (*n*:12) and non-normo-zoospermic (*n*:14) men	Whether the protein profile of extracellular vesicles was related to fertility	Two unique proteins identified in the sEVs of normozoospermic males and three in non-normozoospermic males
Hong et al. (2021) [27]	China	Human	Semen from normo- (*n*:41) and astheno-zoospermic (*n*:42) men	Whether EV piRNAs profile was related to MitoPLD expression and thus to sperm fertility	piRNAs and MitoPLD were reduced in spermatozoa of asthenozoospermic men
Jena et al. (2020) [28]	India	Human	Semen from fertile men (*n*:21) and partners of women with RPL (*n*:21)	Whether seminal EV proteomic profiling was related to RPL	A total of 106 EV proteins were under- and 71 over-expressed in RPL partners. Fifty-six EV proteins were only expressed in RPL partners. This led to defects in paternal gene expression and embryo development
Mei et al. (2019) [29]	China	Human	Semen from men attending infertility center ^1^	Whether seminal EV galectin-3 influenced sperm fertilization capacity	Low levels of seminal EV galectin-3 were associated with low fertilization rates
Vickram et al. (2020) [30]	India	Human	Semen from normo- (*n*:35), oligo-, astheno- (*n*:35), oligoastheno- (*n*:35) and a-zoospermic (*n*:10)	Identify new diagnostic and prognostic biomarkers of infertility among proteins encapsulated in prostasomes	Clusterin protein was differentially expressed between normozoospermic and non-normozoospermic samples
Wang et al. (2021) [31]	China	Mouse	Seminal plasma from young (*n*:3) and older males (*n*:3)	Whether aging affects content of seminal EVs and thus embryo implantation	Aging influenced seminal vesicle size and content. Perfusion of seminal EVs from young males increased implantation rate.

^1^ The number of samples is not indicated. Abbreviations: ARRDC4 (Arrestin Domain Containing 4), EVs (extracellular vesicles), miRNAs (microRNAs), MitoPLD (Mitochondrial cardiolipin hydrolase), piRNAs (piwi interacting RNAs), RPL (recurrent pregnancy loss).

## Data Availability

Not applicable.

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
