# Peer review of "Seminal Extracellular Vesicles and Their Involvement in Male (In)Fertility: A Systematic Review"

_ijms, 2023, doi:10.3390/ijms24054818_

Round 1

Reviewer 1 Report

The Review entitled" Seminal extracellular vesicles and their involvement in male (in)feritility: a systematic review", firstly reported a systematic review about relation between seminal EVs content and sperm fertility potential. Although the review is well written and follow all the procedures for the systematic review and Meta-Analysis (PRISMA 2020), my major concern is about the inclusion criteria used for paper selection. 

Since 2014 EVs characterization is mandatory as reported in the MISEV guidelines (Lötvall J et al. Minimal experimental requirements for definition of extracellular vesicles and their functions: a position statement from the International Society for Extracellular Vesicles. 2014;3:26913. doi: 10.3402/jev.v3.26913. eCollection 2014. Thery C, et al. Minimal information for studies of extracellular vesicles 2018 (MISEV2018): a position statement of the international society for extracellular vesicles and update of the MISEV2014 guidelines. J Extracell Vesicles. 2018;7:1535750) and required a combination of electronic microscopic evaluation and western blot analysis with specific EVs marker antibodies.

No criteria about procedure for characterization was considered in this study and in fact as reported in Table 1. many papers included in this systematic review do not properly characterized the isolated EVs.  

In addition for papers pubblished before MISEV guideline a partials characterization by electron icroscopy is expected as previous reported in many previous key studies (Stegmayr B and RonquistG. Promotive effect on human sperm progressive motility by prostasomes. Urol Res. 1982;10(5):253-7. doi: 10.1007/BF00255932. Harding H, et al. Endocytosis and intracellular processing of transferrin and colloidal gold-transferrin in rat reticulocytes: demonstration of a pathway for receptor shedding. Eur J Cell Biol. 1984;35:256-63. Raposo G, et al. 1996. B lymphocytes secrete antigen-presenting vesicles. J. Exp. Med. 183:1161–72. MacKenzie A, et al. 2001. Rapid secretion of interleukin-1β by microvesicle shedding. Immunity 15:825–35.)

For this reason, I do not recommend this paper for publication in IJMS.

Author Response

We are grateful for the reviewer's comments and criticisms, which undoubtedly are intended to improve the scientific quality of the manuscript. Below, we respond point-by-point to the reviewer's comments or criticisms.

Comments and Suggestions for Authors

The Review entitled" Seminal extracellular vesicles and their involvement in male (in)feritility: a systematic review", firstly reported a systematic review about relation between seminal EVs content and sperm fertility potential. Although the review is well written and follow all the procedures for the systematic review and Meta-Analysis (PRISMA 2020), my major concern is about the inclusion criteria used for paper selection.

Question 1.- Since 2014 EVs characterization is mandatory as reported in the MISEV guidelines (Lötvall J et al. Minimal experimental requirements for definition of extracellular vesicles and their functions: a position statement from the International Society for Extracellular Vesicles. 2014;3:26913. doi: 10.3402/jev.v3.26913. eCollection 2014. Thery C, et al. Minimal information for studies of extracellular vesicles 2018 (MISEV2018): a position statement of the international society for extracellular vesicles and update of the MISEV2014 guidelines. J Extracell Vesicles. 2018;7:1535750) and required a combination of electronic microscopic evaluation and western blot analysis with specific EVs marker antibodies.

Answer 1.- We appreciate the reviewer's reminder of the requirements recommended by ISEV for the characterization of extracellular vesicles. In this regard, it is appropriate to point out that two of the authors of this manuscript are members of ISEV and actively involved in the drafting of the recommendations compiled in MISEV2022 that will be published in 2023. It should also be pointed out that MISEV recommends, but not mandates.

Question 2.- No criteria about procedure for characterization was considered in this study and in fact as reported in Table 1. many papers included in this systematic review do not properly characterized the isolated EVs.

Answer 2.- The present systematic review follows PRISMA guidelines with a clear and well-defined inclusion criterion, namely experimental articles focused on seminal extracellular vesicles that included the word “fertility” or a related word such as “infertility, subfertility, or fertilization” in the title, objective(s), and/or key words. Obviously, another inclusion criterion could have been applied, but the authors consider that the one chosen here is sound and appropriate based in view of the starting hypothesis and the consequent objective proposed in this systematic review. Including MISEV recommendations as another inclusion/exclusion criterion would not have been appropriate, as the MISEV recommendations began in 2014 and many of the included articles were published earlier. In fact, most of the papers that do not provide EV characterization data were published before 2014. The reviewer should understand that articles should not be excluded for not following recommendations that were published after or just at the time of the publication of their article. Moreover, not comply the MISEV recommendations does not preclude the publication of an article focused on extracellular vesicles and, thereby, it should not be a reason to exclude them from a systematic review. In fact, the articles that the reviewer intended to exclude are published in prestigious journals such as Molecular Human Reproduction or Journal of Andrology. We agree with the reviewer about the relevance of isolation and characterization procedures, and for this reason the systematic review summarizes in Table 1 how extracellular vesicles were isolated and characterized in each of the included articles. It should be noted that all the included articles followed proper isolation procedures according to MISEV recommendations, even those published before 2014. Therefore, it is reasonable to consider and assume that the extracellular vesicles were well isolated. Despite this, the discussion of this systematic review makes it clear that the results of articles that do not provide EV characterization data should be treated with caution.

Question 3.- In addition for papers pubblished before MISEV guideline a partials characterization by electron icroscopy is expected as previous reported in many previous key studies (Stegmayr B and RonquistG. Promotive effect on human sperm progressive motility by prostasomes. Urol Res. 1982;10(5):253-7. doi: 10.1007/BF00255932. Harding H, et al. Endocytosis and intracellular processing of transferrin and colloidal gold-transferrin in rat reticulocytes: demonstration of a pathway for receptor shedding. Eur J Cell Biol. 1984;35:256-63. Raposo G, et al. 1996. B lymphocytes secrete antigen-presenting vesicles. J. Exp. Med. 183:1161–72. MacKenzie A, et al. 2001. Rapid secretion of interleukin-1β by microvesicle shedding. Immunity 15:825–35.)

Answer 3.- We thank the reviewer for highlighting articles published prior to 2014 that provided EV characterization data. As active members of the ISEV, we agree with the reviewer in the relevance of complying with the recommendations contained in MISEV guidelines. However, from an objective point of view, we consider that is not advisable to remove articles that does not meet the MISEV recommendations, since, as mentioned above, MISEV recommendations are not mandatory. Moreover, the application of this strict criterion could lead to biased results, and even losing valuable information. Once again, we would like to remind that the discussion in the systematic review points out the articles that do not follow the MISEV recommendations, highlighting that their results should be treated with caution, compared to others.

Question 4.- For this reason, I do not recommend this paper for publication in IJMS.

Answer 4.- We regret the reviewer's recommendation, but we humbly hope that our responses to reviewer’s questions will prompt you to reconsider. The present systematic review provides the scientific community with relevant information on the subject and may be useful in addressing future research.

Reviewer 2 Report

The article is based more on a statistical analysis of the articles that had EV as their subject and seems to miss the objective of the level of information and scientific description of this subject now. -

- I recommend its technical-scientific development with reference to the scientific explanation of Ev so sEV.

The topic has an interesting and future topic, I think that the article once published will open up new horizons of knowledge in this field.

Since it is a review, I think that the topic - EV - should be more detailed at a general level, to inform and enlighten the reader about this topic. The subject is new even for connoisseurs, I think that the introduction deserves to be developed more than the 8 and a half lines (L. 47-55).

Also, lines 56-69 should add information on sEVs with reference to morphometry, morphology, biochemical content, mode of action, effect, possibly supplemented with iconography and references.

I recommend improving the sEVs figures/images, with legends and explicit morphological notes.

Author Response

We are grateful for the reviewer's comments and criticisms, which undoubtedly are intended to improve the scientific quality of the manuscript. Below, we respond point-by-point to the reviewer's comments or criticisms.

Comments and Suggestions for Authors

Question 1.- The article is based more on a statistical analysis of the articles that had EV as their subject and seems to miss the objective of the level of information and scientific description of this subject now.

Answer 1.- The article is a systematic review and, as such, is an objective analysis of the research studies performed on a subject. Specifically, the presents study systematically review the available literature to identify, select, and synthesize all published research articles on the involvement of seminal extracellular vesicles in fertility. The present review follows the PRISMA guidelines and therefore describes the scientific content of only the included articles, which were selected according to inclusion/exclusion criteria defined in a PICO (population, intervention, comparison, outcome) table.

Question 2.- I recommend its technical-scientific development with reference to the scientific explanation of Ev so sEV.

Answer 2.- We have expanded in the introduction the paragraphs aimed at defining the compositional and functional characteristics of nonspecific EVs and seminal EVs following the reviewer's recommendation. This led to the inclusion of new references.

Question 3.- The topic has an interesting and future topic, I think that the article once published will open up new horizons of knowledge in this field.

Answer 3.- We welcome the reviewer's opinion about the relevance of the article for future research in this field, which is still in its infancy.

Question 4.- Since it is a review, I think that the topic - EV - should be more detailed at a general level, to inform and enlighten the reader about this topic. The subject is new even for connoisseurs, I think that the introduction deserves to be developed more than the 8 and a half lines (L. 47-55).

Answer 4.- As per reviewer´s request, we have extended the stated paragraph, providing more information about extracellular vesicles.

Question 5.- Also, lines 56-69 should add information on sEVs with reference to morphometry, morphology, biochemical content, mode of action, effect, possibly supplemented with iconography and references.

Answer 5.- As per reviewer´s request, we have also extended the stated paragraph to provide more information about seminal extracellular vesicles.

Question 6.- I recommend improving the sEVs figures/images, with legends and explicit morphological notes.

Answer 6.- We the authors do not understand what you mean by improving the figures/images of sEVs. We assume you are referring to the drawings of the EVs in Figure 3. Accordingly, we have replaced the original drawing of EVs with a cryo-EM image from our archive showing the heterogeneity of seminal extracellular vesicles. We hope we have done what the reviewer asked.

Round 2

Reviewer 1 Report

I agree with authors that MISEV recommends and not mandates, but half of the articles selected in this review did not reported kind of extracellular characterization or if they were reported were inappropriate (total protein quantification). Given that the subject of the review are the EVs, the EVs characterization by using at least one method is fundamental to understand that EVs are correctly isolate and that the protein or RNA encapsulated in sEVs could be related to male (in)fertility.

Author Response

Comment: I agree with authors that MISEV recommends and not mandates, but half of the articles selected in this review did not reported kind of extracellular characterization or if they were reported were inappropriate (total protein quantification). Given that the subject of the review are the EVs, the EVs characterization by using at least one method is fundamental to understand that EVs are correctly isolate and that the protein or RNA encapsulated in sEVs could be related to male (in)fertility.

Response: We fully agree with the reviewer's statement that it is necessary to adequately characterize EVs and thus demonstrate that they have been isolated correctly. Our manuscript makes this statement very clear in the discussion and we recommend taking with caution the results of manuscripts that have not adequately characterized EVs. However, in this systematic review it would not be correct to discard manuscripts for not having adequately characterized EVs according to the MISEV guidelines, especially if the manuscripts had been published before the MISEV guidelines became available. In this regard, remember that most of the manuscripts that do not adequately characterize EVs were published before the MISEV guidelines were available.